# Frequent Karaoke Training Improves Frontal Executive Cognitive Skills, Tongue Pressure, and Respiratory Function in Elderly People: Pilot Study from a Randomized Controlled Trial

**DOI:** 10.3390/ijerph17041459

**Published:** 2020-02-24

**Authors:** Atsuko Miyazaki, Hayato Mori

**Affiliations:** Technology and Innovation Hub, Cluster for Science, RIKEN, Saitama 351-0198, Japan; morihaya49@gmail.com

**Keywords:** karaoke, elderly, cognitive training, frontal executive functioning, tongue pressure, sarcopenia, respiratory function, randomized controlled trial

## Abstract

We tested whether karaoke training improves cognitive skills and reduces the risk of physical function impairments. We conducted a single-blinded randomized controlled trial in 26 elderly participants at residential care facilities, who were generally healthy or required the lowest level of care. Participants were near the threshold for mild cognitive impairment with the Montreal Cognitive Assessment (MoCA) and close to the sarcopenia cut-off with the skeletal muscle mass index. Pulmonary function as measured with spirometry and tongue strength were used as markers for physical functions affected by sarcopenia. Karaoke training occurred once a week for two hours, with an hour of homework assigned weekly. Karaoke training significantly improved the Frontal Assessment Battery at bedside (FAB) compared with an active control group receiving scratch art training (*F* = 8.04, permutation *p*-value = 0.013). Subscore improved with inhibitory control (*F* = 7.63, permutation *p*-value = 0.015) and sensitivity to interference (*F* = 11.98, permutation *p*-value = 0.001). We observed improved tongue pressure (*F* = 4.49, permutation *p*-value = 0.040) and pulmonary function by a greater increase in FIV1 (*F* = 5.22, permutation *p*-value = 0.047). Engaging elderly people, especially those in care homes, with karaoke training exercises that are moderately physically challenging may be a key to slowing cognitive decline and preventing dysphagia by sarcopenia.

## 1. Introduction

Changes in cognitive ability affect the lives of elderly people. However, the extent of this impact has been reported to be more significant and more negative in those with mild cognitive impairment (MCI) that is not serious enough to interfere with everyday activities [1]. Moreover, individuals with MCI are at an increased risk of developing Alzheimer’s [2] or other forms of dementia [3]. In Japan, it was dementia that the largest number of the reason to obtain a nursing care qualification in 2016. As such, maintaining or improving cognitive function may be important for preventing dementia and the need for further care [4,5,6,7]. 

Recent studies have shown that intervention programs that incorporate cognitive training can improve cognitive function and the plasticity of the nervous system [8,9,10,11]. Among such programs, a unique cognitive training intervention has been developed, referred to as learning therapy (LT) [9,12,13,14]. An LT intervention study reported beneficial effects of the training on a diverse range of cognitive functions related to the inhibitory performance of executive functions, oral episodic memory, focusing, and processing speed in healthy elderly people [12]. LT comprises a learning program for two tasks: reading aloud and solving arithmetic problems. Given the similarity between the reading aloud task of LT and karaoke, we implemented a karaoke training in which participants used a lyric book analogous to the reading task of LT. Based on previous studies of the reading aloud task of LT, we predicted karaoke would elicit similar improvements in frontal executive cognitive skills. Also, it is not just cognitive function, we focused on reading aloud because long periods of speech or a loud speech requires the physical activity [15].

Elderly people are also more likely to suffer from sarcopenia, a decline in physical functions due to decreased muscle strength. This is especially true in hospitalized patients, in whom muscular disuse-related atrophy and malnutrition may lead to loss of muscle mass [16]. Dysphagia is known to occur when generalized sarcopenia is present [16,17]. Tongue pressure declines as skeletal muscles weaken with age, and this affects swallowing ability [18,19]. Indeed, decreased tongue pressure represents a symptom of dysphagia [20]. Even in healthy people, tongue strength has a large impact on bolus clearance [19,21,22] and an inability to regularly consume food implicates a decrease in tongue pressure [23]. Moreover, maximum tongue pressure correlates with body mass index (BMI) and arm muscle area [24]. It has further been reported that lower tongue pressure correlates with decreased hand grip strength and jumping ability [25].

Previous research thus suggests that a decrease in tongue pressure is part of the sarcopenia syndrome [24], and that sarcopenia involves the swallowing muscles [18]. In addition to aspiration pneumonia and death, maintaining sufficient muscle strength to prevent dysphagia is an important concern in elderly people. Exercise has been shown to be effective for strengthening the tongue and improving dysphagia and functional swallowing in healthy elderly people [26,27,28,29]. The availability of the simple tongue pressure test has led to attempts at early prevention of sarcopenia and frailty due to swallowing dysfunction [23]. In Japan, tongue pressure evaluation was introduced during a medical service fee revision in 2016. Further investigation of the possibility of prevention and improvement of dysphagia caused by sarcopenia with training that targets tongue pressure and the respiratory muscles is necessary.

During aging, the chest wall stiffens, and the elastic recoil of the lungs decreases and restricts [30]. Respiratory muscle contraction is necessary to move the chest wall and lungs [15] and is required to create enough force for phonation. Moreover, respiratory system is equally important for both phonation and swallowing function [31,32]. 

The respiratory muscles are skeletal muscles that are also affected by sarcopenia, and a relationship has been reported between respiratory muscle strength and conventional sarcopenic indices [33,34]. In cervical spinal cord injury (SCI), the maintenance of respiratory muscle function is particularly important and training that encompasses singing appears to have positive effects on respiratory function, as reported in a randomized-controlled trial [35]. Moreover, patients with chronic obstructive pulmonary disease (COPD) have benefited from singing, with the approach spreading and patients with respiratory diseases joining singing groups in the United Kingdom [36]. Singing appears to lead to improvements in forced expiratory volume in the first second (FEV1) and forced vital capacity (FVC) [37,38,39]. Finally, the effects of singing on the respiratory system have led to its use as an adjuvant therapy in pulmonary rehabilitation [40,41].

Karaoke is a singing activity that requires long phonation, and it has become a popular interactive leisure activity for many people worldwide following the development of the karaoke machine in Japan in the 1970s. Satoh et al. [42] investigated the influence of karaoke training and voice training on cognitive function in 10 patients with Alzheimer’s disease. The results suggested improved processing speed during spatial cognitive function tasks following six months of the intervention, with further support for improvements from neuroimaging data. This study was conducted to verify whether karaoke training prevents cognitive decline in a group of elderly individuals who were generally healthy or required the lowest level of care. Based on previous studies of reading aloud and speech therapy, we predicted similar effects of karaoke. As karaoke training requires an increase in activity of the respiratory and tongue muscles for an extended period, we also hypothesized that karaoke training improves physical function, especially respiratory function, due to an exercise effect. Furthermore, we measured tongue strength as a proxy for sarcopenic dysphagia, as tongue pressure becomes weaker with age and tongue strength is susceptible to whole body sarcopenia [19]. Although this measure is not an accurate reflection of overall swallowing function, sufficient evidence indicates that tongue pressure is a convenient potential indicator of sarcopenia due to swallowing dysfunction [23]. In a recent study, tongue pressure training not only increased tongue pressure but also improved oral diadochokinesis [43]. Training the movement of the tongue with frequent karaoke should, therefore, strengthen motor neurons of the tongue muscles and increase tongue pressure. As tongue pressure is related to cognitive function [20,44,45,46], and since both measures are susceptible to aging and neurological diseases, an inspection of the potential beneficial effects of karaoke is essential.

The aim of this study was to first assess the feasibility of frequent karaoke training in elderly people who are healthy or require the minimum amount of care, and then observe the effects of such regular karaoke practice on the participants’ cognitive and physical functions. We, therefore, designed a single-blinded randomized controlled trial (RCT) at two care facilities in Japan.

## 2. Materials and Methods

### 2.1. Trial Design and Setting 

This RCT was registered in the University Hospital Medical Information Network (UMIN) Clinical Trial Registry (UMIN000031189) and was conducted in Nerima-Ku, Tokyo, Japan. Written informed consent to participate in the study was obtained from each participant before enrollment and all study procedures adhered to the Declaration of Helsinki. The study protocol and informed consent form were approved by the ethics committee of RIKEN in Saitama (ref. Wako3 29−7).

We conducted a single-blinded RCT with two parallel groups: a karaoke training group (Intervention group) and a scratch art group (Active control group). Members of the groups were assigned randomly irrespective of their total Montreal Cognitive Assessment (MoCA) [47] scores. Research personnel who assessed cognitive and physical functions were blinded to the group assignment. The trainer did not know details about the participants’ backgrounds or psychological profiles, and they did not have specific psychological aims. The Consolidated Standards of Reporting Trials (CONSORT) statement [48] (http://www.consort-statement.org/home/) was used as a framework. Figure 1 presents the trial design.

### 2.2. Participants

Participants were recruited via posters and briefing sessions about the study provided to residents of two care homes in Nerima-Ku, Tokyo. These care homes are residential facilities for elderly people aged 65 years and older, in particular elderly people with a low income and difficulties living at home because of physical problems, their environment, or their housing circumstances [49]. Given the health of the residents, while they cannot live independently, they require the lowest level of care regarding their Activities of Daily Living (ADLs) [50] and Instrumental Activities of Daily Living (IADLs) [51]. The care homes provide communal housing with a private room, meals, and recreation for the residents. If a resident requires care for certain ADL areas such as assistance with bathing, care is provided by an external service.

Care home residents were included in the study if they were identified as native Japanese speakers with no diseases known to affect the central nervous system, heart, or respiratory system. Moreover, none of the participants were involved in other cognitive and exercise-related intervention studies. Furthermore, they did not participate in regular exercise consisting of more than 1 hour per week at the gym, nor were they involved in exercise therapy as part of medical rehabilitation. Participants were excluded if they had a vision and/or hearing impairment. 

Out of 70 residents living at the two care homes, 28 participants were recruited, among whom 21% were using external care services. Of all included participants who provided written informed consent, two were excluded because they did not meet the inclusion criteria (due to severe Meniere’s disease and lacking the ability to understand the tasks). To obtain information on participants’ cognitive functions, we used the MoCA [47], which has high sensitivity and specificity, to detect MCI, as a cognitive screening tool with a cut-off value of 26. Randomization into the intervention or control group was conducted after the participants provided informed consent (*n* = 26).

### 2.3. Sample Size

The sample size was not pre-determined for this pilot study, as no previous studies have conducted an intervention test for cognitive and physical function levels with karaoke.

### 2.4. Overview of Interventions

Each program (karaoke training for the intervention group or scratch art for the control group) was delivered for 12 weeks, on 3 scheduled days each week. Participants were asked to attend once a week, and they had to attend at least nine of the 12 sessions to be included in the analysis. On scheduled days, completion criteria were a note of practice content for karaoke training and a completed artwork for the scratch art class, which was assigned as homework 1 week before the scheduled day. Although there were individual differences, both programs took about 2 hours per week.

Measures of cognitive and physical function were taken before the start and after the end of the program. Participants also completed a series of cognitive and physical function tests before beginning the program schedule (Pre-test). After 12 weeks, participants were re-examined using the same neuropsychological and behavioral tests (Post-test).

### 2.5. Karaoke Training (Intervention Group)

Full details of the karaoke training have been provided in a previous publication [42]. The karaoke training program ran for 1 hour, once a week for 12 consecutive weeks on weekday afternoons at the care home, and homework of 1 hour was given each week. 

Usually, karaoke involves a music video and lyrics displayed on a monitor, the participant sings into a microphone, and the voice is mixed with background music. The karaoke monitor displays lyrics in time with the music, and the color of the words changes to indicate the timing when singing along. Since such karaoke machines are very large and expensive, it has proven difficult for elderly people to own one personally or use one daily. To overcome these challenges, we used the small and portable Karaoke Book (developed by PDMHD Co., Ltd., Tokyo, Japan) for daily karaoke training. We implemented the karaoke training in a way that enabled us to compare the results to LT programs: lyrics were provided to participants in a lyrics book, similar to the reading aloud task in LT, and phonetic timing was thus not determined by any visual cues.

For both the session and the homework, the handy karaoke device was used that was provided to each participant at no cost. The karaoke machine with the microphone and adapter measured 3.3 × 18 × 20 cm^3^ (H × W × L) and was lightweight at 700 g, making it easy for the elderly to carry. The karaoke machine was accompanied by a 48-song lyrics book that corresponds to the songs built into the machine. The selected songs were songs that elderly people favored when they were young or were very well-known popular songs from recent years. Participants entered the song number from the lyrics book into the karaoke machine, pressed the play button, and sang the song with a microphone while looking at the lyrics book.

The karaoke training sessions were guided by a professional singer in groups of four or five people. The participants were informed that the purpose of the training was to sing as many songs as possible. Each session consisted of three sections. The first section brushed up on songs practiced during the previous week. In the next section, participants chose their favorite songs from the lyrics book and sang in groups. The selected songs were adapted to the participant’s pitch range by the karaoke machine. In the last section, new songs to be mastered were chosen. New songs were proposed as homework, which were to be practiced 3 times a week, for 20–60 min per week, and described in a weekly report. As the sessions continued, participants increased the number of songs that they could sing and increased the difficulty level of the songs they chose. 

### 2.6. Scratch Art (Active Control Group)

For the control group in behavioral research, an active control group is preferred to a non-treated control group [52]. Participants residing in care homes will also experience social effects from participating in an intervention program [53]. Therefore, we used an active control group to control for the number of social contacts in both groups. The karaoke group received a multi-faceted extensive intervention program, which includes a combination of two or more interventions, including the preparation of procedures and the use of the tool [54]. Therefore, the active control program was designed to control for positive effects that could be attributed to participating in this intervention study. Control participants performed scratch art [55]. 

Scratch art uses coated paper with a black surface. Participants can draw on the paper by scratching at the surface with a stick to reveal various colors. As part of the active control program, participants were provided with one piece of scratch art paper per week. Scratch art sessions were held at the care home once a week for 12 weeks on consecutive weekday afternoons and lasted half an hour. Homework was to complete one artwork per week. Participants were asked to draw with the trainer at the scratch art session, and to complete and submit the drawing as homework the following week.

### 2.7. Cognitive Function Measures

To evaluate the beneficial effects of karaoke training on cognitive function, we comprehensively measured cognitive function with the MoCA [47] and the Frontal Assessment Battery at bedside (FAB) [56] at before and after intervention (Table 1). These tests were chosen because we wanted to verify whether karaoke had the same effect on cognitive improvement as LT. The MoCA was used to assess the degree of cognitive function of participants and the FAB was used to assess training effectiveness. 

The MoCA is a 10-min 30-item cognitive screening tool with high sensitivity and specificity to detect MCI within the normal range of the Mini Mental State Examination (MMSE). One point is added to account for individuals with an educational history of 12 years or less. The total score was from 0 to 30, with a cut-off of 26; scores lower than that indicate general cognitive dysfunction. 

As in previous LT studies, we also used the FAB. The FAB is a 10-min 18-point instrument presented on a screen to assess executive cognitive functions. FAB items include Similarities (conceptualization), Lexical Fluency (mental flexibility), Motor Series (programming), Conflicting Instructions (sensitivity to interference), Go No-Go (inhibitory control), and Prehension Behavior (environmental autonomy). 

### 2.8. Physical Function Measures 

To evaluate the beneficial effects of karaoke training on physical functions, we measured tongue pressure and pulmonary function at before and after intervention (Table 1). 

Tongue pressure was measured with a JMS tongue pressure manometer (TPM−01, JMS Co. Ltd., Hiroshima, Japan) [20,28,57]. Participants were seated and asked to place the balloon in their mouth and on their tongue. The balloon is inflated with air at a pressure of 19.6 kilopascals (kPa), which is calibrated as 0. A researcher confirmed that the probe was in the correct position. Participants were then asked to compress the balloon with their tongue against the hard palate of their mouth as hard as possible for 7 seconds. The maximum value of three trials was recorded in kPa (from −9.9 to 100.0). The average value of tongue presser of elderly people who are residing in a senior welfare facility is less than 30 kPa, which is considerably lower than those who stay at their own homes [23]. Lower values indicate greater degrees of swallowing dysfunction. Tongue pressure assessment is covered by Japanese government medical insurance policy since 2016.

Pulmonary function was measured using spirometry with the MIR Spirobank G (Medical International Research, Rome, Italy). The measuring device was equipped with a mouthpiece fitted with a disposable turbine that was replaced for each participant. Participants were seated and asked to hold the mouthpiece with one hand while using a nose clip to prevent the loss of air volume through the nose. Participants were asked to seal their lips around the mouthpiece, breathe in fully until their lungs were absolutely full, and then blast the air out as far as possible with maximum effort until the lungs were completely empty. Measurements were compared with standard values from Japanese people according to the Japan Respiratory Society (JRS). The following spirometry parameters were assessed: FVC, forced inspiratory volume in the first second (FIV1), and FEV1. These evaluations were completed by independent evaluators who were experienced physical therapists.

### 2.9. Psychological Measures

Several psychological measures were assessed before and after the intervention. We used the Geriatric Depression Scale (GDS) [58] and the Life Satisfaction Index-K (LSI-K) [59] to assess quality of life (QOL) (Table 1).

### 2.10. Other and Sarcopenia Measures

Several background measures were recorded, including age, sex, educational history (less than/more than 12 years), height (cm), and weight (kg). We used the Barthel ADL index (ADL) [50] and the IADL scale [51], administered in interview form. 

Bioelectrical impedance analysis (BIA) using the InBody S10 (Biospace Co., Ltd., Seoul, Korea) was used to measure body weight (kg), BMI (kg/m^2^), and the skeletal muscle mass index (SMI; kg/m^2^). The SMI is a value obtained by dividing the muscle mass of the limb by the height (m^2^) and is a measure of both muscle mass and sarcopenia [60]. At the Asian Working Group for Sarcopenia (AWGS) 2014, the SMI cutoff values recommended for elderly people over 65 years of age were 7.0 kg/m^2^ for men and 5.7 kg/m^2^ for women using BIA values [61]. Lower values indicate greater degrees of low muscle mass and sarcopenia. These evaluations were performed by independent evaluators who were experienced physical therapists (Table 1).

### 2.11. Statistical Analyses

The primary outcome measure of the present intervention was FAB score because based on previous studies of the reading aloud task of LT [12,14], we predicted karaoke would elicit similar improvements in frontal executive cognitive skills. To check group differences, we calculated score changes (post-intervention score minus pre-intervention score) for all cognitive, physical, and psychological measures. We conducted an analysis of covariance (ANCOVA) with permutation tests for each score change because it is applicable to small samples and corrects for the occurrence of false positives. An ANCOVA with permutation tests was conducted to determine significant differences in score changes between the intervention and control groups. Therefore, the independent variable was group (karaoke intervention group vs. scratch art active control group) and score changes were the dependent variable. Several factors that might have affected the outcome of the study, such as cognitive and psychological measures at baseline, sex, and age, as indicated in LT studies [12,14], were therefore added as covariates. Moreover, as physical function affects body size, physical function at baseline, sex, age, height, and weight were also included as covariates. 

The “aovp” function of the lmPerm package was used for all ANCOVAs with permutation tests [62], it is suitable for verifying the effectiveness of intervention tests with small sample size [14,63,64]. Moreover, effect sizes (*η*^2^) were calculated by the sum of squares between the groups and the sum of squares of the ANCOVA permutation test with eta squared (*η*^2^) [65]. Missing values were not included in the analyses. Statistical significance was set at *p* < 0.05, and all analyses were conducted with R version 3.4.3 (R Core Development Team, 2008, Vienna, Austria).

## 3. Results

### 3.1. Background Characteristics

Between April 2018 and May 2018, 28 participants were enrolled in the study, two of whom were excluded at baseline because they did not meet the inclusion criteria. Of the final 26 participants (mean age = 82.43 years; range = 69–93 years), 78.26% were women and 56.52% had more than 12 years of education. Women had an average SMI of 5.79 kg/m^2^ (SD = 0.84; range = 4.1 kg/m^2^–7.4 kg/m^2^; AWGS cutoff value for sarcopenia, 5.8 kg/m^2^). Men exceeded the AWGS cutoff value of 7.0 kg/m^2^ with an average of 7.56 kg/m^2^ (SD = 1.34; range = 5.9 kg/m^2^–9.4 kg/m^2^. 

The average MoCA score for all participants was 24.42 (SD = 4.13; range = 12.00–30.00) and 57% of participants scored 25 points or less. Therefore, these indexes suggested participants were in the sarcopenia and MCI zones of these measures.

Following their providing of informed consent, 14 participants were randomly assigned to the intervention group and 10 to the control group. There were no significant differences in any baseline measures between the intervention and control groups (all *p* > 0.09, Table 1). During the 12 weeks, 23 of the 26 participants completed all measures and the training protocol, while one intervention and two control participants discontinued the study. Reasons for drop-out were refusal to complete measurements at follow-up (two controls) and negative experiences with the training (one intervention) (Figure 1). To test for the training effects of karaoke on cognitive and physical functions, we conducted ANCOVA permutation tests for score changes in all measures (Table 2).

### 3.2. Cognitive Function 

To test the effects of karaoke on cognitive executive function skills, we performed ANCOVAs with permutation tests to assess changes in MoCA and FAB total score (Table 2). We observed significant improvements in total FAB scores in the intervention group (mean = 0.77; SD = 1.97) compared with the control group (mean = −1.80; SD = 2.44) (F(1, 18) = 8.04, *p* = 0.011; permutation *p*-value = 0.013; *η*^2^ = 0.31) (Table 2). There was no significant difference in total MoCA scores between the groups (intervention mean = 0.92, SD = 1.30 vs. control mean = 0.00, SD = 3.02; F(1, 18) = 0.43, *p* = 0.523; permutation *p*-value = 0.667; *η*^2^ = 0.02) (Table 2). 

To demonstrate an improvement in cognitive function indicated by FAB scores, we present pre-intervention scores in Table 3 and score changes for each subscore of the FAB in Table 4. There were improved scores for Conflicting Instructions (sensitivity to interference) in the intervention group compared with the control group (intervention mean = 0.46, SD = 0.84 vs. control mean = −0.30, SD = 1.10; F(1, 18) = 11.98, *p* = 0.003; permutation *p*-value = 0.001; *η*^2^ = 0.41) (Table 4). Moreover, for Go No-Go (inhibitory control) scores, there was a significant improvement in the intervention group (mean = 0.69, SD = 1.07) compared with the control group (mean = 0.10, SD = 1.30) (F(1, 18) = 7.63, *p* = 0.013; permutation *p*-value = 0.015; *η*^2^ = 0.26) (Table 4). 

In contrast, there was no significant difference between the intervention and control groups in Similarities (conceptualization) scores (intervention mean = −0.31, SD = 0.61 vs. control mean = 0.20, SD = 0.60; F(1, 18) = 2.45, *p* = 0.135; permutation *p*-value = 0.158; *η*^2^ = 0.15), Lexical Fluency (mental flexibility) scores (intervention mean = 0.31, SD = 0.46 vs. control mean = −0.50, SD = 1.20; F(1, 18) = 2.96, *p* = 0.103; permutation *p*-value = 0.144; *η*^2^ = 0.13), Motor Series (programming) scores (intervention mean = −0.31, SD = 0.82 vs. control mean = −1.30, SD = 0.90; F(1, 18) = 2.78, *p* = 0.113; permutation *p*-value = 0.108; *η*^2^ = 0.13), or Prehension Behavior (environmental autonomy) scores (intervention mean = −0.08, SD = 0.27 vs. control mean = 0.00, SD = 0.00; F(1, 18) = 0.55, *p* = 0.469; permutation *p*-value = 0.824; *η*^2^ = 0.06).

### 3.3. Physical Function

We observed improved tongue pressure in the intervention group (mean = 3.71 SD = 5.96) compared with the control group (mean = −1.55, SD = 3.40) (F(1, 16) = 4.49, *p* = 0.05; permutation *p*-value = 0.040; *η*^2^ = 0.22) (Table 2). Moreover, pulmonary function improved in the intervention group indicated by a greater increase in FIV1 (mean = 0.18, SD = 0.37) compared with the control group (mean = −0.29, SD = 0.42) (F(1, 16) = 5.22, *p* = 0.036; permutation *p*-value = 0.047; *η*^2^ = 0.25) (Table 2). 

However, there was no significant difference in FVC between the intervention and control groups (intervention mean = 0.26, SD = 0.30 vs. control mean = −0.04, SD = 0.31; F(1, 16) = 2.06, *p* = 0.170; permutation *p*-value = 0.163; *η*^2^ = 0.11). Furthermore, there was no significant difference in FEV1 between the groups (intervention mean = 0.04, SD = 0.37 vs. control mean = 0.01, SD = 0.37; F(1, 16) = 0.43, *p* = 0.521; permutation *p*-value = 0.667; *η*^2^ = 0.03).

### 3.4. Psychological Measures

Psychological measures did not differ significantly before and after the intervention. There was no significant difference in GDS between the intervention and control groups (intervention mean = −0.77, SD = 2.91 vs. control mean = −2.00, SD = 3.19; F(1, 18) = 0.00, *p* = 0.965; permutation *p*-value = 1.000; *η*^2^ = 0.00). There was no significant difference in LSI-K between the intervention and control groups (intervention mean = 0.54, SD = 1.50 vs. control mean = 0.10, SD = 2.26; F(1, 18) = 1.94, *p* = 0.181; permutation *p*-value = 0.233; *η*^2^ = 0.01) (Table 2). 

### 3.5. Summary 

Our results suggest that primary outcome measures improved based on total FAB scores. This indicates that frequent karaoke training led to improved performance on Go No-Go (inhibitory control) and Conflicting Instructions (sensitivity to interference) as measures of cognitive function, as well as tongue pressure and FIV1.

## 4. Discussion

In this study, we investigated the effects of frequent karaoke training on cognitive and respiratory function and tongue pressure in residents of a care home. The participants were healthy elderly people requiring the lowest level of care who reached cut-off thresholds for MCI and sarcopenia (Table 1). Our intervention program was well received at the care homes and continued for 3 months. The results of the RCT clearly indicate that karaoke training led to improved cognitive and physical function compared with an active control group. Improvements in cognitive function were indicated by changes in total FAB scores (Table 2) as well as in subscores for Conflicting Instructions (sensitivity to interference) and Go No-Go (inhibitory control) (Table 4). Significant improvements in physical function were also demonstrated by changes in tongue pressure and pulmonary function (FIV1), which are susceptible to aging and sarcopenia and provided a proxy measure of sarcopenic dysphagia. These results support our hypothesis that karaoke intervention improves cognitive and sarcopenic dysphagia as well as respiratory function in the elderly (Table 2).

Karaoke training appears to promote similar improvements in cognitive function as reading aloud does in LT. Previous studies of LT in individuals with dementia [9] and in healthy elderly people enrolled in an RCT for 6 months [13] have reported improved FAB scores. The FAB provides an executive function measure by assessing different frontal lobe functions within six subsets. Studies have also reported improvements in FAB subscores including Similarities (conceptualization) [9], Lexical Fluency (mental flexibility), and Conflicting Instructions (sensitivity to interference) [13]. In the current study, karaoke training for 3 months led to total FAB score improvements similar to those reported in LT studies. Moreover, we observed significant increases in FAB subscores for Go No-Go (inhibitory control) with an effect size of 0.29 and Conflicting Instructions (sensitivity to interference) with an effect size of 0.41 in the karaoke training group. Therefore, we suggest that karaoke training has effects that are similar to those of LT, by improving Conflicting Instructions (sensitivity to interference) scores. 

The sensitivity to interference task requires participants to show behavioral self-regulation while following verbal commands and refrain from following visual information [56]. Reading in LT uses a combination of several cognitive processes such as visually presented word recognition, conversion from a graphical representation of a word to a phonological expression, analysis of the meaning of words, control of pronunciation, and working memory [12]. Singing words from a lyrics book requires a combination of more cognitive processes than reading aloud. We suggest that this explains why karaoke training had an additional effect on the Go No-Go (inhibitory control) process that has not been found for LT. The inhibitory control task elicits a false-alarm motor response that the participant should inhibit for inappropriate responses [56]. Response inhibition is required for accurate phonation timing and the high level of executive control required to sing songs. In particular, music training can improve inhibitory control abilities [66], because of the more robust domain-independent transfer effects that are present compared with specific memory and cognitive training [67]. Physical, especially aerobic, exercise is recommended (grade B) in clinical practice guidelines to prevent dementia [68]. Physical activity protects gray and white matter loss and reduces neurotoxic factors [69]. However, these guidelines do not provide exact instructions on the amount of exercise. Elderly people have lower aerobic capacity and reduced motor functions compared with younger adults. Indeed, according to the recent Physical Activity Guidelines for Americans (PAGA), on an absolute scale, light exercise intensity may represent moderate or severe intensity for older people [70]. The Compendium of Physical Activities provides a classification system that standardizes the Metabolic Equivalent (MET) intensity levels of different physical activities and singing songs while sitting is included in the list. Therefore, frequent karaoke training may provide an exercise effect for elderly people in residential care who are inclined to physical inactivity. Singing requires the use of respiratory muscles and control of breath while seated is rated as a 1.5 MET intensity level of physical activity, i.e., light MET. It is very light exercise for healthy people but not for patients with SCI and COPD, and elderly people with sarcopenia who have lower aerobic capacity and motor function. In SCI and COPD, training including singing has been reported to improve respiratory function [35,36]. In sarcopenia, not only body skeletal muscle loss but also muscle decomposition is accelerated in the tongue and diaphragm, and atrophy has been observed in respiratory muscles [71]. However, sarcopenia has been suggested to be reversible with exercise, including head-raising and other muscle strengthening exercises. As respiratory strength increases, the chest wall can move with greater force, which reduces chest stiffness [72]. Chest wall mobility is correlated with lung capacity, and chest expansion is known to be related to maximum inspiratory volume. Therefore, physical activity requiring the use of respiratory muscles and control of breath in karaoke training may exert an exercise load on the respiratory system. 

In the current study, we tested this hypothesis by observing effects on respiratory muscle strength with forced inspiratory and expiratory volumes [73]. Our results suggest improved pulmonary function indicated by increased FIV1 values. Moreover, a relatively large effect size (*η*^2^ = 0.25) was observed for our intervention in terms of FIV1 improvement (Table 2). Indeed, both inspiratory volume and tidal volume increased, suggesting that respiratory function had improved. Moreover, increased ventilation expands the range of motion of the chest wall [15]. Therefore, our results suggest that karaoke training improves physical function related to respiration and exerts an exercise effect. 

Previous research suggests that long durations of speech or loud speech recruit active expiratory muscles [15], but in the current study, expiratory volume was not affected. Regardless of age and depending on the degree of comprehension of the task, participants can maximize inspiration, while effortfully increasing expiration appears to be difficult [74]. In addition, since FIV1 decreases due to muscle weakness [75], it has also been suggested that FEV1 is easier to improve by training than other respiratory functions [76,77], which explains why karaoke training may easily lead to an inspiratory training effect.

We observed that tongue pressure improved with karaoke training, with a relatively large effect size (*η*^2^ = 0.22). Participants at baseline were near the SMI cutoff value indicating sarcopenia, and tongue pressure was also below the standard value (Table 1). Tongue pressure is related to body strength. Indeed, recent studies have shown a relationship between tongue pressure and sarcopenia in the elderly. However, the availability of the simple tongue pressure test has led to the early prevention of sarcopenia and frailty resulting from swallowing dysfunction [23]. Maeda and Akagi [78] suggest that sarcopenia is an independent risk factor of dysphagia among older people in hospital. Moreover, a frail body and tongue are related to a decline in swallowing function, as well as dysphagia caused by sarcopenia. Butler et al. [79] describe an association between aspiration status and both tongue and handgrip strength in healthy elderly people. Swallowing function is complex, but despite the lack of a direct measure, early prevention of sarcopenia and frailty due to swallowing dysfunction has been enabled by the simple tongue pressure test [23]. 

Because declining swallowing function leads to nutrition and hydration deficits, as well as aspiration pneumonia and death, maintaining enough muscle strength to prevent dysphagia is an important concern in elderly people. According to data from a survey by the Ministry of Health, Labor and Welfare, there were approximately 97,000 cases of pneumonia and 36,000 cases of aspiration pneumonia in 2017, together representing the third leading cause of death in Japan (https://www.mhlw.go.jp/toukei/saikin/hw/jinkou/geppo/nengai17/dl/kekka.pdf). Therefore, maintaining or improving tongue pressure is very important in aging societies. For the prevention of sarcopenia dysphagia, training that presses the tongue against the palate increases the strength and volume of the tongue [27,43,80]. Recent research has also reported that the dexterity of the tongue improved simultaneously [43]. Tongue dexterity ratings represent oral diadochokinesis and are used to evaluate oral motor skills. In Japan, the “PATAKARA Exercise” (/pa/, /ta/, /ka/, /ra/) is used to create tongue movements and improve oral diadochokinesis to maintain swallowing function in the elderly [81,82]. Therefore, even in individuals with sarcopenia, karaoke training is considered to enhance motor neurons associated with the tongue, which results in an increase in tongue pressure.

All cognitive and physical function indicators measured in this study are related and show a decline in aging [83,84,85,86,87,88,89,90,91]. There are strong correlations between tongue pressure and MMSE scores [20,45], and between pulmonary function and MMSE scores [44,46]. Moreover, swallowing problems occur early in the course of neurologic disease and dementia [92,93]. The elderly are immediately impaired in these functions, particularly when exposed to additional stressors such as acute illness, hospitals, surgery, or chemoradiotherapy. Therefore, a combination of intervention programs for these functions may maintain and improve the quality of life for the elderly. 

In terms of psychological measures, we observed no improvement in QOL or GDS scores following karaoke training. The current study was conducted in two care homes that provide housing, while residents live life freely and care levels are low. Therefore, a 3-month intervention period might not have been sufficient to lead to enhancements in psychological measures such as QOL.

The “Creative Aging” movement, which creates relationships for the elderly through art activities, is increasing in popularity worldwide [94,95,96,97]. The movement places emphasis on the importance of different processes involved in art activities and enables participants to take on different transformative roles regardless of age. For example, in the current study, a young trainer learns old songs from the elderly and the art experience creates a new relationship for the elderly participant. Although doubt remains regarding the effectiveness of karaoke, the intervention investigated here shows clear effects on cognition and physical function. Moreover, it represents a familiar experience for elderly people in Japan and provides opportunities for improving human relationships. Furthermore, participants in the intervention group completed 3 months of the intervention, with only one person dropping out. These findings suggest that karaoke may provide a new effective intervention as part of the “Creative Aging” movement.

The current study has some limitations. Importantly, the number of participants that were available for inclusion was limited. Although the involved care homes have a joint capacity of 70 people, residents are able to go to the gym and work freely. Therefore, only a limited number of elderly people agreed with the schedule of the intervention. Moreover, to observe effects at an elderly facility, a no-treatment control group may be necessary, in addition to an active control group; here, however, we focused on an active control group to control for the number of social contacts in both groups of participants. In the previous LT study, improved scores for verbal tasks of the FAB were observed for Similarities (conceptualization) and Lexical Fluency (mental flexibility). In the conceptualization task, participants must conceptualize two words. Meanwhile, the mental flexibility task is a task of verbal fluency and word production [56]. It is possible that karaoke training for 6 months (the same duration as used in the LT study) might have also led to improvements in these capabilities. Moreover, in future studies, a broader range of cognitive tests should be used to elucidate beneficial effects similar to those of LT across a wider range of cognitive functions.

## 5. Conclusions

In this study, we report on improvements in cognitive function following frequent karaoke training in a sample of elderly residents at a care home. Our results suggest that karaoke training has similar effects on cognitive function as LT, with improved inhibitory performance and executive functions. This is not unexpected because similar effects have already been seen with reading aloud in LT. Furthermore, because karaoke training had an influence on physical functions, it led to an exercise effect on respiratory function and tongue pressure. Respiratory function and tongue pressure reflect swallowing function affected by sarcopenia in elderly people, improvements of which can enhance QOL.

Given the relationship between cognitive and swallowing functions, as well as their association with mortality and neurological diseases, we suggest that karaoke training provides participants with a fun, arts-based intervention that can improve well-being as part of the “Creative Aging” movement.

## Figures and Tables

**Figure 1 ijerph-17-01459-f001:**
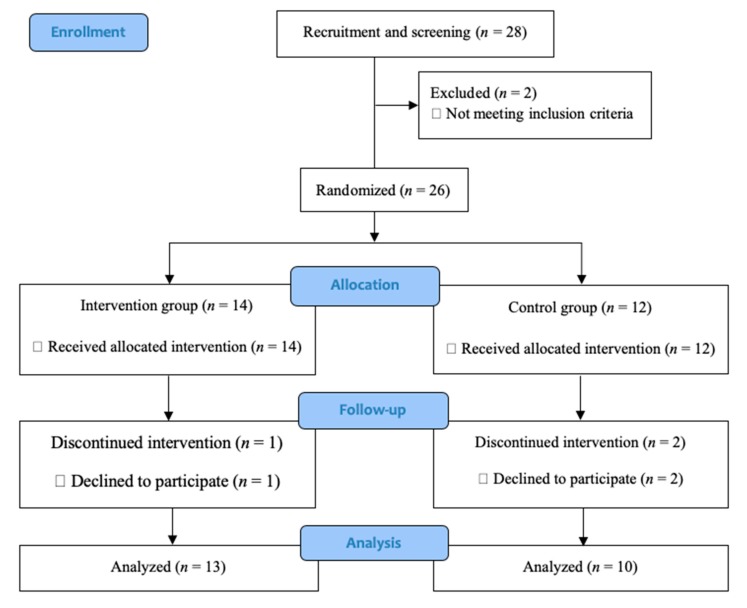
Consolidated standards of reporting trial (CONSORT) diagram.

**Table 1 ijerph-17-01459-t001:** Participant characteristics at baseline.

Criterion	Intervention Group		Control Group		*T*-Test *p*-Value
	(*n* = 14)		(*n* = 12)		
	Mean (SD)	Range	Mean (SD)	Range	
Age, years	80.00 (7.46)	70 to 93	83.58 (7.18)	69 to 93	0.23
Sex (% female)	71.43		83.33		0.49
Education (% with >12 years)	28.57		60.00		0.28
ADL score	84.11 (19.75)	40 to 100	94.79 (8.36)	75 to 100	0.09
IADL score	99.64 (1.34)	95 to 100	96.67 (6.15)	80 to 100	0.09
MoCA score	24.57 (3.39)	19 to 30	24.25 (5.01)	12 to 30	0.85
FAB total score	13.14 (3.42)	6 to 17	12.92 (3.23)	8 to 17	0.86
GDS score	4.36 (2.71)	0 to 8	6.08 (3.12)	1 to 12	0.14
LSI-K score	5.00 (1.84)	3 to 8	3.83 (1.99)	1 to 7	0.13
Tongue pressure (kPa)	27.12 (8.08)	16.70 to 42.25	26.45 (6.78)	13.10 to 34.30	0.82
FVC (liter)	1.85 (0.58)	0.65 to 2.80	1.76 (0.58)	0.66 to 2.66	0.68
FEV1 (liter)	1.42 (0.48)	0.56 to 2.27	1.30 (0.29)	0.81 to 1.88	0.48
FIV1 (liter)	0.49 (0.41)	0.00 to 1.06	0.71 (0.46)	0.00 to 1.25	0.20
Body height (cm)	155.00 (12.60)	131.0 to 178.0	149.90 (8.01)	140.0 to 169.0	0.24
Body weight (kg)	57.84 (15.81)	33.6 to 86.7	49.54 (8.49)	33.80 to 61.20	0.12
BMI (kg/m^2^)	23.73 (4.02)	16.7 to 30.7	22.08 (3.77)	16.30 to 29.90	0.29
SMI (kg/m^2^)	6.45 (1.40)	4.1 to 9.4	5.81 (0.81)	4.4 to 7.4	0.18

Values given are mean (standard deviation) unless stated otherwise. SD, standard deviation; ADL, Activity of Daily Living; IADL, Instrumental Activity of Daily Living; MoCA, Montreal Cognitive Assessment; FAB, Frontal Assessment Battery at bedside; GDS, Geriatric Depression Scale; LSI-K, Life Satisfaction Index-K; FVC, forced vital capacity; FEV1, forced expiratory volume in the first second; FIV1, forced inspiratory volume in the first second; BMI, body mass index; SMI, skeletal muscle mass index.

**Table 2 ijerph-17-01459-t002:** Comparison of score changes after intervention.

	Intervention Group (*n* = 13)		Control Group (*n* = 10)		ANCOVA *p*-Value	Permutation *p*-Value	Effect Size (*η*^2^)
	Mean	SD	Mean	SD			
MoCA score	0.92	1.30	0.00	3.02	0.523	0.667	0.02
FAB total score	0.77	1.97	–1.80	2.44	0.011	0.013 *	0.31
GDS score	–0.77	2.91	–2.00	3.19	0.965	1.000	0.00
LSI-K score	0.54	1.50	0.10	2.26	0.181	0.233	0.01
Tongue pressure (kPa)	3.71	5.96	–1.55	3.40	0.050	0.040 *	0.22
FVC (liter)	0.26	0.30	–0.04	0.31	0.170	0.163	0.11
FEV1 (liter)	0.04	0.37	0.01	0.37	0.521	0.667	0.03
FIV1 (liter)	0.18	0.37	–0.29	0.42	0.036	0.047 *	0.25

**p* < 0.05. SD, standard deviation; Score changes, post-intervention score minus pre-intervention score; MoCA, Montreal Cognitive Assessment; FAB, Frontal Assessment Battery at bedside; GDS, Geriatric Depression Scale; LSI-K, Life Satisfaction Index-K; FVC, forced vital capacity; FEV1, forced expiratory volume in the first second; FIV1, forced inspiratory volume in the first second.

**Table 3 ijerph-17-01459-t003:** Frontal Assessment Battery (FAB) subscores in the intervention and control groups at baseline.

	Intervention Group (*n* = 14)		Control Group (*n* = 12)		T-Test *p*-Value
	Mean(SD)	Range	Mean(SD	Range	
Similarities (conceptualization)	1.57 (1.02)	0 to 3	1.17 (0.83)	0 to 2	0.280
Lexical Fluency (mental flexibility)	2.07 (0.92)	1 to 3	2.50 (0.67)	1 to 3	0.194
Motor Series (programming)	2.64 (0.63)	1 to 3	2.92 (0.29)	2 to 3	0.181
Conflicting Instructions (sensitivity to interference)	2.21 (0.80)	1 to 3	1.92 (1.31)	0 to 3	0.485
Go No-Go (inhibitory control)	1.64 (1.28)	0 to 3	1.42 (1.31)	0 to 3	0.661
Prehension Behavior (environmental autonomy)	3.00 (0.00)	3	3.00 (0.00)	3	1.000

FAB, Frontal Assessment Battery at bedside; SD, standard deviation.

**Table 4 ijerph-17-01459-t004:** Comparison of score changes for FAB subscores after intervention.

	Intervention Group (*n* = 13)		Control Group (*n* = 10)		ANCOVA *p*-Value	Permutation *p*-Value	Effect Size (*η*^2^)
	Mean	SD	Mean	SD			
Similarities (conceptualization)	−0.31	0.61	0.20	0.60	0.135	0.158	0.15
Lexical Fluency (mental flexibility)	0.31	0.46	−0.50	1.20	0.103	0.144	0.13
Motor Series (programming)	−0.31	0.82	−1.30	0.90	0.113	0.108	0.13
Conflicting Instructions (sensitivity to interference)	0.46	0.84	−0.30	1.10	0.003	0.001 *	0.41
Go No-Go (inhibitory control)	0.69	1.07	0.10	1.30	0.013	0.015 *	0.29
Prehension Behavior (environmental autonomy)	−0.08	0.27	0.00	0.00	0.469	0.824	0.06

**p* < 0.05. FAB, Frontal Assessment Battery at bedside; SD, standard deviation; Score changes, post-intervention score minus pre-intervention score.

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
