# Peer review of "Frequent Karaoke Training Improves Frontal Executive Cognitive Skills, Tongue Pressure, and Respiratory Function in Elderly People: Pilot Study from a Randomized Controlled Trial"

_ijerph, 2020, doi:10.3390/ijerph17041459_

Round 1

Reviewer 1 Report

The paper by Miyazaki and Mori reports on the use of karaoke in a sample of elderly people to assess cognitive skills, tongue strength, and respiratory function. The study was conducted in three months of karaoke training. The authors report improvements in tongue strength and some parameters of respiratory function concluding that karaoke leads to improved executive function and physical function as evidenced by improved respiratory parameters and tongue strength.

This paper is written well and the study has been executed well. I only have minor suggestions for improving some sentences in the manuscript.

In the entire text, change ‘single-blind’ to ‘single-blinded’. Remove hyphenation in line 14. In line 30, ‘have an impact on’ reads as understatement. Please change this to ‘affects’. Rephrase the statement in line 34 to remove the long string of nouns acting as adjective (nursing care requirement authorization) and change word choice regarding the usage of ‘cause’. One would wonder why ‘dementia’ cause ‘authorization’. In line 48, delete ‘recruitment of’. In line 50, change the second occurrence of ‘muscle’ to ‘muscular’. In line 55, change ‘indicates’ to ‘implicates’. In line 71, use ‘respiratory system is’ instead of ‘systems are’. In line 95, change ‘become’ to ‘becomes’. In line 97, ‘there is sufficient evidence to indicate’ to ‘sufficient evidence indicates’. Change the sentence starting in line 116 to, ‘We conducted a single-blinded RCT with two parallel groups: a karaoke training group (Intervention group) and a scratch art group (Active control group).’ Members of the groups were assigned randomly irrespective of their total Montreal Cognitive Assessment (MoCA) scores.’ If this is not intended, please rephrase for clarity. In line 125, delete ‘from a care home’. In line 211, remove ‘pre- and post-intervention periods’. Use ‘before and after intervention’. Period does not make sense because end-points were measured, not throughout the period of intervention. As for comment 13, see line 226: use ‘before’ and ‘after’. In line 228, delete ‘Measurements of maximum tongue pressure were performed with a measuring device’ and replace with ‘The device was’. The sentence ending in 240 sounds repetitive. Please revise and remove repetition. At the end of line 244, change ‘measured’ to ‘air’. In line 280, delete ‘and’. In line 320, write ‘post-intervention score minus pre-intervention score’ instead of ‘post score minus pre score’. In line 510, move ‘LT’ before the word ‘study’. In line 522, remove the comma after ‘unexpected’.

Author Response

We wish to thank the reviewer for those comments.
Those errors have been corrected in accordance with the reviewer's comment.
I send the revised manuscript to the reviewer. Our new revision uses Microsoft Word's "track changes" feature. Please see the attachment.

Reviewer 2 Report

The paper is well written and follow the guidelines.

The abstract does not report results.

Author Response

Thanks to reviewers for pointing out the lack of results in the abstract.
Following this, we have added and corrected including the results.

Abstract: We tested whether karaoke training improves cognitive skills and reduces the risk of physical function impairments. We conducted a single-blinded randomized controlled trial in 26 elderly participants at residential care facilities, who were generally healthy or required the lowest level of care. Participants were near the threshold for mild cognitive impairment with the Montreal Cognitive Assessment (MoCA) and close to the sarcopenia cut-off with the skeletal muscle mass index. Pulmonary function as measured with spirometry and tongue strength were used as markers for physical functions affected by sarcopenia. Karaoke training occurred once a week for two hours, with an hour of homework assigned weekly.  Karaoke training significantly improved the Frontal Assessment Battery at bedside (FAB) compared with an active control group receiving scratch art training (F = 8.04, permutation p-value = 0.013). Subscore improved with inhibitory control (F = 7.63, permutation p-value = 0.015) and sensitivity to interference (F = 11.98, permutation p-value = 0.001). We observed improved tongue pressure (F = 4.49, permutation p-value = 0.040) and pulmonary function by a greater increase in FIV1  (F = 5.22, permutation p-value = 0.047) .  Engaging elderly people, especially those in care homes, with karaoke training that are moderately physically challenging may be a key to slowing cognitive decline and preventing dysphagia by sarcopenia.

(Words: 198)

Reviewer 3 Report

Review of the ijerph-729219 manuscript

The authors tested whether karaoke training improves cognitive skills and reduces the risk of 11 physical function impairments in elderly people. The manuscript analyzes the results of a single-blinded randomized 12 controlled trial in 26 elderly participants.

The paper is well-written and organized. A comprehensive literature review is proved, and the methods and statistical analyses are described properly. Comprehensive experiences are conducted, and sufficient discussions are provided. Therefore, I would like to suggest accepting it in the current format.

Author Response

We wish to express our strong appreciation to the reviewers on our paper. We have had the manuscript rewritten by an experienced scientific editor, who has improved the grammar and stylistic expression of the paper. If the calibration certificate on English is required, I will send it to IJERPH.
Also, we will report to you that this paper has been revised to English and in response to comments from other reviewers.
We send the latest version, please see the attachment.
